# An Analysis of Participation and Performance of 2067 100-km Ultra-Marathons Worldwide

**DOI:** 10.3390/ijerph18020362

**Published:** 2021-01-06

**Authors:** Angelika Stöhr, Pantelis Theodoros Nikolaidis, Elias Villiger, Caio Victor Sousa, Volker Scheer, Lee Hill, Beat Knechtle

**Affiliations:** 1Medbase St. Gallen Am Vadianplatz, 9001 St. Gallen, Switzerland; angelika.stoehr@bluewin.ch; 2Exercise Physiology Laboratory, 18450 Nikaia, Greece; pademil@hotmail.com; 3School of Health and Caring Sciences, University of West Attica, 12243 Athens, Greece; 4Institute of Primary Care, University Hospital Zurich, 8091 Zurich, Switzerland; evilliger@gmail.com; 5Bouve College of Health Sciences, Northeastern University, Boston, MA 02115, USA; cvsousa89@gmail.com; 6Ultra Sports Science Foundation, 69130 Pierre-Bénite, France; v.scheer@ultrasportsscience.org; 7Department of Pediatrics, Division of Gastroenterology & Nutrition, McMaster University, Hamilton, ON L8N 3Z5, Canada; hilll14@mcmaster.ca

**Keywords:** ultra-marathon, participation, performance gap, male–female differences

## Abstract

This study aimed to analyze the number of successful finishers and the performance of the athletes in 100-km ultra-marathons worldwide. A total of 2067 100-km ultra-marathon races with 369,969 men and 69,668 women competing between 1960 and 2019 were analyzed, including the number of successful finishers, age, sex, and running speed. The results showed a strong increase in the number of running events as well as a strong increase in the number of participants in the 100-km ultra-marathons worldwide. The performance gap disappeared in athletes older than 60 years. Nevertheless, the running speed of athletes over 70 years has improved every decade. In contrast, the performance gap among the top three athletes remains persistent over all decades (F = 83.4, *p* < 0.001; _p_η^2^ = 0.039). The performance gap between the sexes is not significant in the youngest age groups (20–29 years) and the oldest age groups (>90 years) among recreational athletes and among top-three athletes over 70 years. In summary, especially for older athletes, a 100-km ultra-marathon competition shows an increasing number of opponents and a stronger performance challenge. This will certainly be of interest for coaches and athletes in the future, both from a scientific and sporting point of view.

## 1. Introduction

In recent years, ultra-endurance running has experienced exponential growth in popularity, with more organized events each year [1]. From 1998 to 2010, North America’s number of events increased from 21 to 52, respectively [2]. An ultra-marathon is defined as any running race that takes longer than 6 h or is longer than a marathon distance (42.195 km) [3]. Depending on the specific event, ultra-marathons can take on different formats, including time-limited events [4] or distance-limited events [1,5].

Trends in participation and performance in ultra-marathon running have been investigated in several studies [1,3,6,7,8,9]. The popularity of ultra-marathons [8,10] as well as the number of ultra-marathons [1] has increased worldwide in the last decades. Evaluations in several ultra-marathon races worldwide showed an exponential increase in the number of participants [1,8]. This exponential increase of participation in ultra-marathons worldwide, on the one hand, depends on male participation [8]. On the other hand, it depends on the increased number of women who reached a plateau of up to 20% in the last decades [1,6,11]. Furthermore, most ultra-marathon participants are master athletes, who are defined as athletes over 35 years of age who undergo systematic training for competitions [6,10,11]. Nevertheless, even children and adolescents under the ages of 19 years are participating in 100-km events, which have also seen an increase in participation, especially over the last couple of decades [9,12]. 

However, two studies on participation analysis of ultra-marathons showed that there was a decline in the number of participants in these individual events. The number of participants in single ultra-marathons like the “100-km Lauf von Biel”, as the oldest 100-km ultra-marathon in the world [9], or the “Comrades Marathon” primarily increased in the 1970s and 1980s. Thereafter, a continuous decline occurred [9,13]. The initial increase and the subsequent decrease of participation depended primarily on the number of male participants [9]. However, the worldwide trend of 100-km ultra-marathons from 1998 to 2011 shows an increasing number of participants [8]. This contrary development of the participants in ultra-marathons in particular single events and the overall view worldwide highlights the importance of examining participants worldwide with a more extensive database to better understand the trends of athletes running ultra-marathons. Additionally, there is currently only data on the number of participants for other distances and regions, such as the 161 km (100 m) ultra-marathons in North America [1]. The available information on the development of the number of participants in 100-km ultra-marathons is limited to a short period of investigation, from 1998 to 2011 [8]. This study aimed to close this gap by a worldwide data collection of participation in 100-km ultra-marathons over several decades. Furthermore, our study aims to investigate the worldwide trend of 100 km ultra-marathon participation despite changing organizers and events.

Furthermore, the performance of the ultra-marathon participants is an important aspect for both athletes and scientists. An expected result in various studies was the faster performance in male age group runners than in female age group runners [3,14,15,16]. Nevertheless, an important study result showed that female and male runners’ performance gap decreased or remained stable in ultra-marathons in the last decades as well as in older age groups [3,14,17]. This performance gap in older age groups was smaller in 161-km (100 m) ultra-marathons than in 80-km (50 m) races [14]. However, the analysis of the gap in performance in all races held worldwide in single ultra-marathon distance such as 100-km is missing, especially regarding the trend across years and for master athletes.

Furthermore, the performance trends showed different results depending on the type of the athlete (i.e., elite or recreational level) [7]. Over the past decades, the top men and top women runners of 100-km ultra-marathons became faster [7]. However, no further improvement had been shown in the last years for the top runners [3,6,7]. The performance gap between male and female elite runners and between top-ten runners got smaller in the last decades in 100-km ultra-marathons [7]. Therefore, a further analysis of differences in performance of top runners in older age groups for 100-km ultra-marathons is relevant worldwide and is missing so far.

Significant improvements in the performance of master athletes in other disciplines, such as the 100-m run, the 400-m middle distance run, and the 100-m freestyle swimming, have already been demonstrated [18]. The extent of the performance improvement was more significant in the older age groups, especially among women [18]. Similarly, an increase in performance was shown in master triathletes over 40 years of age in swimming, cycling, and running [19]. Among marathon runners, a quarter of marathon runners aged 60 years exceeded more than half of the younger runners [20]. In the “New York City Marathon”, the master athletes improved their running time more than the younger athletes, whose performance plateau has been reached [21]. Up to now, there is no data collection for the performance change of older athletes in a 100-km ultra-marathon.

Therefore, the aim of this study was primarily for the 100-km ultra-marathon, firstly, to examine the number of participants worldwide with a more extensive database and also to include the missing last years since 2011 in the analysis, secondly, to put the increasing number of participants in relation to the increasing number of events, thirdly, to examine the performance based on age, especially among master athletes, fourthly, to analyze the change in performance over the last decades, and, fifthly, to highlight the performance gap between the sexes. 

Based upon the findings as mentioned above regarding participation and performance trends for ultra-marathon races held worldwide, we hypothesized (i) an increase in participation worldwide in the last decades, (ii) the participants/event ratio would be declining, (iii) no further significant performance gap in older age groups would occur, (iv) an increase in performance among older athletes over the past decades, and (v) men were faster in most age groups, but not for younger age groups (i.e., younger than the age of peak ultra-marathon performance) and older age groups (i.e., in the oldest age groups). These assumptions would be interesting for athletes who plan their ultra-marathon career and for scientists who explore athletes’ efficiency.

## 2. Methods

### 2.1. Ethical Procedures

This study was approved by the Institutional Review Board of Kanton St. Gallen, Switzerland, with a waiver of the participants’ requirement for informed consent as the study involved the analysis of publicly available data (Ethical Committee St. Gallen 01–06–2010). The study was conducted in accordance with the recognized ethical standards outlined in the Declaration of Helsinki (2013).

### 2.2. Data Sampling

Data were obtained from the website of DUV (Deutsche Ultramarathon Vereinigung) from their database using a python script [22]. From this publicly accessible results database, for all 100-km ultra-marathons since the first official race in 1960 to 2019, the first name, last name, sex, age, nationality, and race time of each successful finisher were recorded. The entire sample included data from 2067 events with a total of 715,571 participants. Of these, 13.4% were women and 86.6% were men who ran a 100-km ultra-marathon between 1960 and 2019, which were documented on the DUV website (http://statistik.d-u-v.org/geteventlist.php). The age of the participants was between 18 and 94 years. The average age was 43.8 ± 10.6 years. The nationality of the participants showed a distribution of 10.0% Germans, 10.1% French, and 79.9% other nationalities. The race time over 100 km was between 363.9 and 5221.9 min, with an average race time of 854.5 min (standard deviation (SD) 255.2 min). 

### 2.3. Statistical Analysis

The Shapiro-Wilk and Levene’s tests were applied for normality and homogeneity, respectively. General linear models were as follows: average speed (three-way analysis of variance (ANOVA)) and sex × decade × age group, and average running speed considering only the top three athletes per event (two-way ANOVA) and sex × decade. Sex was always included as a fixed factor, and decade and age group were included as random factors. Partial eta square (_p_η^2^) was calculated for each model and used as a measure of effect size considering small ≥ 0.01, medium ≥ 0.06, and large ≥ 0.14. When interactions were found (*p* < 0.05), pairwise comparisons were applied to identify the differences more accurately. The Mauchly test verified the hypothesis of sphericity, and when violated, the degrees of freedom were corrected by the Greenhouse–Geisser estimates. An independent t-test was used to test the difference of age between sex, considering only the top three athletes. Significance level was set at 5% (*p* < 0.05). All statistical procedures were performed using the Statistical Package for the Social Sciences (SPSS version 26. IMB, Armonk, NY, USA) and GraphPad Prism (version 8.4.2. GraphPad Software LLC, San Diego, CA, USA).

## 3. Results

The total sample comprehended data from 2067 events, from 1960 to 2019, and was clustered by decades. The number of events was found to be significantly higher (>35.0%) in the last decade (2010–2019), growing from 379 in 2000–2009, to 1373 (Figure 1A). The number of participants also had an exponential increase in the last decade in both men (34.3%) and women (44.9%), with 369,969 men and 69,668 women enrolling in 100-km ultra-marathons between 2010 and 2019 (Figure 1B). The ratio of participants/event significantly decreased from 1960 to 1989, reaching a plateau of 320 participants per event from 2000 to 2019 (Figure 1C). All age groups showed an increased participation throughout the years, but those with more participants from 2000 to 2019 were aged between 30- and 59-years-old (Figure 1D). Interestingly, participants with 80 or more years of age have increased their participation from zero, in the first decade, to 100 in 2010–2019.

The ηmultifactorial model for performance showed significant effects for sex (F = 15.2, *p* = 0.005; _p_η^2^ = 0.656), age groups (F = 8.7; *p* < 0.001; _p_η^2^ = 0.622), decade (F = 13.7, *p* < 0.001; _p_η^2^ = 0.773), and interactions: sex × decade (F = 4.3, *p* = 0.001; _p_η^2^ = 0.040) and age group × decade (F = 23.8; *p* < 0.001; _p_η^2^ = 0.963). Pairwise comparisons showed that performance significantly (*p* < 0.001) increased up to the decade 1990–1999, with no further significant (*p* > 0.05) differences afterwards (Figure 2A). The average running speed of men was higher than women up to 1989 (*p* < 0.001), with no further significant differences thereafter (Figure 2A). Age groups 40–49 and 50–59 years were found to run significantly faster than others, completing races in less time (Figure 2B). 

Considering only the top three athletes, performance showed significant effects for sex (F = 12.2, *p* = 0.018; _p_η^2^ = 0.589), decade (F = 14.6, *p* = 0.005; _p_η^2^ = 0.789), and interaction sex × decade (F = 83.4, *p* < 0.001; _p_η^2^ = 0.039). Pairwise comparisons showed that men were faster than women across all decades, and performance significantly increased up to 1999 and decreased in the next two decades (Figure 2C).

All age groups up to 69-years-old increased their performance up to 1999, reaching a plateau or decreasing in subsequent decades. However, only the age groups 70–79 or ≥80-years-old showed an increase in their performance every decade (Figure 3). 

The average age of men ranking in the top three was 39 ± 8 years, with a higher number of athletes between 35- and 42-years-old (Figure 4A). For the women, the average age of athletes ranking in the top three was 40 ± 8 years, with a higher number of athletes between 35- and 43-years-old (Figure 4B). There was a significant difference between the sexes and the average age of the top three finishers (T = 10.4, *p* < 0.001, IC95% (confidence interval) = 0.8–1.2).

When all women and men per age group were considered, men were faster than women in age groups 30–39 to 80–89 years, but not in 20–29 and 90–99 years (Table 1). When the top three women and men per age group were analyzed, men were faster than women in age groups 30–39 to 60–69 years, but not in 20–29, 70–79, and 80–89 years (Table 2).

## 4. Discussion

This study examined the participation and performance trends in 100-km ultra-marathons events from around the world over a 59-year period. This includes 2067 events over a period of 1960–2019. Our aim was to test the hypothesis, first, that the number of participants increased in the last decades. Secondly, that the ratio of participants/event decreased in the last decades. Thirdly, that the performance gap in older age groups of participants would get smaller and, fourthly, that older athletes, especially older women, get faster in the last decades. The main results were (i) the number of participants worldwide in 100-km ultra-marathons increased, (ii) the ratio of participants/event reached a plateau in the last decade after a decline, (iii) there were no further significant performance gaps in age groups over 60 years or older, (iv) older age groups over 70 years increased their performance every decade, and (v) men were faster in most age groups, but not for younger age groups (20–29 years) and older age groups (90–99 years for all and 70–79 and 80–89 years for top three finishers).

### 4.1. Increased Number of Finishers

The first finding in our study was the increasing number of participants in both sexes from 1960 to 2019, especially in 2009–2019. This confirmed our first hypothesis of increasing participation numbers in 100-km ultra-marathons worldwide. Another study also examined the number of participants—in particular, the number of successful finishers—in 100-km ultra-marathon races worldwide over a shorter time period from 1998 to 2011. During this decade, an exponential increase in the number of successful finishers worldwide was found [8]. Similar findings, with an exponential increase of participation, were reported for the fastest runners in 100-km ultra-marathons worldwide [7]. An additional study, investigating participation trends in all 161-km ultra-marathon races in North America between 1977 and 2008, also reported an exponential increase in the number of finishers during the last three decades [1]. The number of participants in other ultra-marathons also increased, e.g., in the evaluation of worldwide data from 24 h ultra-marathons [11] and of worldwide data in 6, 12, 24, 48, 72 h, and 6-day (144 h) and 10-day (240 h) ultra-marathons [9,15].

The increase in participants in our study depends mostly on a higher number of male participants. Nevertheless, in our study, the number of female participants has grown in the last decade from 2010 to 2019. A previous study of 100-km ultra-marathons worldwide between 1998 to 2011 showed similar results that the number of participants increased exponentially for both sexes, and the percentage of 15% women participants remained stable [8]. In contrast, a study of all North American ultra-marathons showed an increase in participants, especially in an increasing number of female athletes [1].

A further finding was that mostly the age group of 30–59 years increased the number of participants. A similar development occurred in the “Comrades Marathon” with an increase in finishers that arises from a higher number of male participants in the age group 30–60 years [13,23]. Additionally, a new age group appeared in 2010 in our study, with participants over 80 years beginning to take part more frequently in these events. Before 2010, there were nearly no participants over 70 years old. Only in the last decade was there a significant number of participants over 70 years old, so that now older athletes have also got opponents of the same age group in competition. The number of older athletes also increased the number of participants in the WSER (Western States 100 m Endurance Run) [6]. Another ultra-marathon evaluation also showed an increasing number of older and younger athletes [24]. So, we can assume that the increasing number of participants was caused by the increasing male as female participants in age group 30–59 years and as well by the new age group of over 80-year-old participants.

### 4.2. Ratio Participants/Event

In contrast to the increasing number of participants worldwide, a decline of participation in particular single events could be observed [3,9,13]. For example, the decrease of participants in single events can be shown in the oldest 100-km ultra-marathon in the world, the “100-km Lauf Biel” [3], or in the “Comrades Marathon” [13]. This decline of participants in particular single events could be a result of an increasing number of events. An increase in events could also be shown in other long-distance events such as the marathons in China [25]. In our study, such a strong increase of 100-km ultra-marathon events could be demonstrated, especially in the last decade from 2010 to 2019. In this last decade of an increasing number of events and increasing participation, the ratio between participants and events was stable. In previous decades up to 1989, the ratio between participants and events decreased. This leads to the assumption that in previous decades before 1989, there was a redistribution of the participants to the events. So, the number of participants in single events declined as the number of events offered increased. 

### 4.3. Performance Gap Disappeared in Older Age Groups 

In accordance with our third hypothesis: no further significant performance gap in older age groups, the third finding in our study was that the significant performance gap disappeared in athletes older than 60 years. Age groups 40–49 and 50–59 years were found to run significantly faster than others, completing races in less time.

In contrast, athletes running shorter distances, such as the New York City Marathon, show a significant decline in performance among elite marathon runners aged 36 years and older [26]. Similarly, the analysis of several marathon races showed a significant performance gap between the age groups of athletes over 56 years in recreational runners [27]. Also, in mountain running, a significant performance gap was observed from the age of 50 years onwards [28]. But, in our study, with an extensive data collection for 100-km ultra-marathon events, no performance difference could be found after the age of 60 years.

As a cause of endurance performance in shorter endurance races such as mountain running for athletes up to 49 years of age, oxygen uptake in the anaerobic zone has already been identified as a major determinant [28]. This oxygen uptake in the anaerobic zone was found to be stable in athletes up to the age of 49 and could be the cause of the previously relatively stable performance of these athletes [28]. Other physiological causes for the endurance performance in older age groups can be body fat and training characteristics [29], sarcopenia [30], or muscle mass and strength [31]. Furthermore, it has been shown that lifelong endurance training can maintain muscle mass and function in the elderly [32]. Similarly, aerobic fitness per unit of body mass could be a factor in the endurance performance, which showed a correlation with the level of competition in ultra-long-distance runners [33]. The second ventilation threshold was also the best aerobic fitness variable [33]. These physiological variables, such as the second ventilation threshold, aerobic fitness per body unit, or muscle mass, could also be used for further studies to investigate the endurance performance of older athletes.

### 4.4. Performance Gap Disappeared Since 1990

Furthermore, there was no more significant performance gap between male and female participants since 1990. Similar results can be shown in shorter 80-km ultra-marathons, where a decrease of performance gap began in 1985 [14]. In an analysis of ultra-marathons of varying lengths, women were also able to reduce the performance gap to men in 6, 72, 144, and 240 h races since 1975 [15], in 24 h races since 1998 [11], and over all decades in 50-km ultra-marathons worldwide [25].

### 4.5. Top Runners’ Performance Gap Remains

In contrast, the top three male athletes are faster than female athletes across all decades. The performance of both sexes increased up to 1999 and decreased in the next decades. However, the performance gap remains permanent, although each year the gap is reducing between the sexes. The top three male athletes remain significantly faster than the top three female athletes, although the performance gap gets smaller over the last decades. The decline in the performance gap can also be shown in a worldwide study of the fastest female and male 100-km ultra-marathoners. In this study, the performance gap decreased from 56% in 1965 to 16% in 2012 [7]. Also, the performance gap of the top ten runners decreased from 46% in 1975 to 14% in 2012 in 100-km ultra-marathons worldwide [7] and in 24 h ultra-marathons worldwide [11].

### 4.6. Older Athletes Get Faster in Last Decades

Our study’s fourth finding was the increasing performance in older age groups over 70 years every decade. So far, performance improvements in older athletes have only been seen in other sports, such as 100-m running, 400-m running, and 100-m freestyle swimming [18], in master triathletes in swimming, cycling, and running [19], and in marathon runners [20,21]. The increasing performance in athletes older than 70 years could be attributed to the increase in general participation rates, especially in the older age groups, who are new. In the last decade, the older age groups might have more experience than the older age groups in previous decades. Such an increase in master athletes’ performance due to increased participation was also observed in other sports like 100-m, 400-m running, and 100-m freestyle swimming [18] and “New York City Marathon” [21,34]. The increasing performance in running speed could also be seen among the younger participants in the ultra-marathon’s early decades. First, running speed improved until 1990 and then deteriorated over the next two decades. This suggests that the master runners have probably not yet reached their performance limits in the ultra-marathon like the younger athletes who have reached a performance plateau. Also, in the “New York City Marathon”, master athletes showed an increase in running speed over the last decades, whereas the younger athletes had reached their performance level and their performance remained unchanged over the last decades [34]. Therefore, with increasing numbers of participants in the older age groups, running speed development would have to be reassessed in the coming decades. The question is when the performance level of the older athletes will be reached in the next decades.

Another aspect that could influence the increase in performance in older ages is physical activity over the entire life span. It could be assumed that older athletes were already very active in sports decades before and that the increasing optimization of training among older athletes leads to an improvement or maintenance of physical function and health. It has already been shown that lifelong physical activity can help to mitigate the loss of many of the characteristics impaired by aging, especially through aerobic and resistance training [35]. Similarly, maintaining training intensity and volume into old age can affect the rate of age-related decrease in VO_2_max and muscle mass in master endurance athletes [36]. It was also shown that the training economy and training intensity decreases less with increasing age. This means that a high level of VO_2_max can be maintained even at an advanced age. Thus, a high training stimulus can limit the decrease in endurance performance even in older athletes. Therefore, master athletes seem to be able to maintain their physical performance with advancing age due to ultra-endurance sports [37]. Similar effects were shown at international swimming competitions, where the number of races per year and age proved to be a success factor [38].

### 4.7. Younger and Older Women Achieve a Similar Performance to Their Male Counterparts

A last important finding was that the youngest (i.e., 20–29 years) and the oldest (i.e., 90–99 years for all and 70–79 and 80–89 years for the top three) women achieved similar performance levels to men. Another study showed a similar development with a decreasing performance gap in older age groups in 161-km races and 80-km races [14]. There were no significant differences in performance from the age of 75 years in 80-km races and from the age of 60 years in 161-km races [14]. This means that the longer the competition’s running distance, the earlier the performance balance between men and women [39]. This trend of a smaller performance gap between men and women in the older than in the younger age groups could also be demonstrated in big data samplings of 50-km ultra-marathons worldwide [40], in single races like “The Comrades Marathon” [13,41], or in the “New York City Marathon” [42]. These findings of the performance gap decline in older age groups in ultra-marathons have already been demonstrated in other sports, such as master swimmers, who no longer have a performance gap in the age group 80–89 years [30]. 

However, in these very young and very old age groups, the number of participants is very low, especially over very long distances like 80-km or 161-km races [14]. Therefore, the result for these very young and very old age groups could have been random [14]. This problem of a very low number of participants in ultramarathon evaluations also became apparent in a study of 6, 12, 24, 48, and 72 h, and 6-day (144 h) and 10-day (240 h) ultra-marathons [15]. A performance gap was even found to increase with age, but this is most likely due to the very small number of women in the older age groups [15]. 

Our study also shows very low participant numbers for these very young age groups under 29 years and the very old age groups over 70 years. Thus, the performance approximations between the sexes of the athletes under 29 and over 70 years of age in our study could also be random. In the middle age groups of our study, a very large number of participants could be included, therefore the significant performance difference between the sexes in the middle age groups is highly valid. Such significant performance differences between men and women were also demonstrated in sprint and endurance distances in sports such as running, swimming, cycling, and triathlon [43]. There has already been shown a stable performance difference between the sexes of 10% [43]. Similarly, a longitudinal study showed a stable performance difference between the sexes in walking speed from the age of 55 years [44]. Another study of running distances up to 200-km showed an average running speed difference of 12.4% between men and women [45]. Whereby, also in this study, the problem of the very low percentage of women in long-distance competitions is pointed out [45]. Therefore, the lack of performance gap between men and women might also be random in our study and difficult to assess conclusively due to the small number of participants in these age groups.

## 5. Limitations and Future Research Lines

Future studies should include more participant characteristics such as aerobic fitness per unit of body, muscle mass, fat mass or years of training. Despite the large amount of data available on the DUV website, these characteristics cannot be retrospectively recorded in our study. Therefore, the gap in the study can only be closed by using a different study design to identify the origin of the data observed in our study in the older age group of ultra-marathon athletes. Another strength of this study is the low number of participants in the higher age groups despite the high volume of data. This showed a trend between age and performance, but the result could be overestimated in the higher age groups. 

Another limitation of the study is the lack of continuity of events, so that in the data collection, events sometimes occur only for short periods of time. This makes it difficult to establish a structural equation model to identify multifactorial dependency relationships. Establishing an event/participant ratio is also difficult because only one event occurred permanently during the period of data analysis. The other events passed and failed during the period of data analysis. Implications for future studies would therefore be to analyze the event/participant ratio within a shorter period of time, e.g., the next decade.

Further measures should be taken with regard to the growing older age groups, as with the higher age exists the risk of medical events with the so far existing standards. Adapted medical assessment for the older athletes would be useful in avoiding medical events such as sudden cardiac death. For coaches and older athletes preparing for an ultramarathon, specific training plans and nutrition plans for the elderly athletes’ requirements are of new importance as well and should be newly developed and adapted for this older age group.

## 6. Conclusions

Based on a large data analysis of over 2076 competitions from 1960 to 2019, a strong growth in the number of participants as well as a significant growth in the number of events offered in the 100-km ultra-marathon worldwide is evident. This initially resulted in a decrease in the participant/event ratio and then led to a plateau in the ratio over the last three decades. An analysis of the older age groups shows the new increasing age group of the over 80-year-old participants. In addition, there is a relatively stable endurance performance in the older age group of the over 60-year-old athletes, which is either due to the lower number of participants or physiological causes. Likewise, the increase in participants’ performance over 70 years of age over the last decades could depend on both above-mentioned factors. Or, the new increasing age group of the over 70-year-old participants may have benefited from the experiences of the previous decades. Further analyses of the growing age group of older participants could provide information on the development of performance and sex differences in old age. For coaches and master athletes, this study shows that despite increasing age, an increase in performance can still be achieved in the next decades and that in the future, the number of participants of older athletes will increase. According to this, there is still a lot of potential in the training economy and training volume of older athletes.

## Figures and Tables

**Figure 1 ijerph-18-00362-f001:**
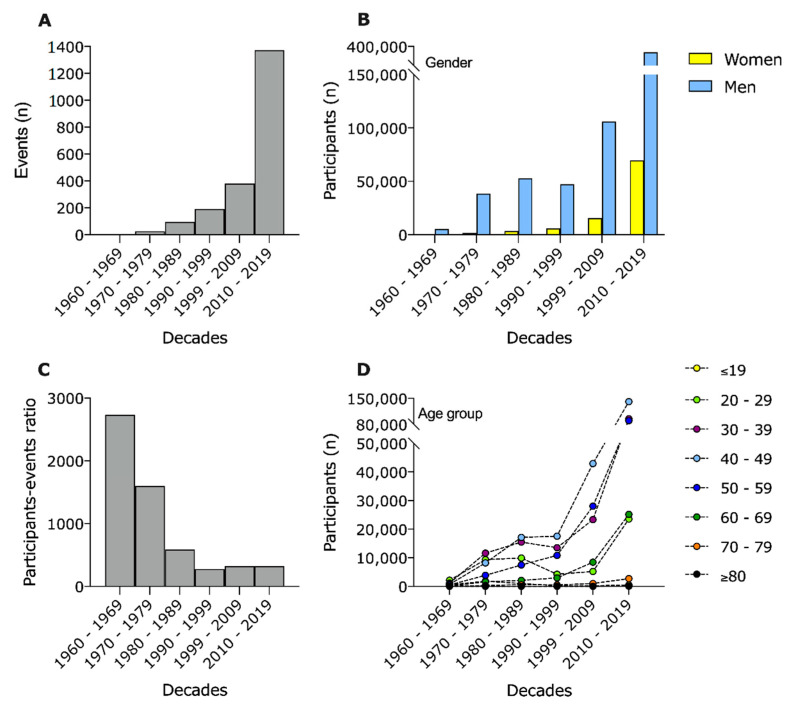
Number of events (total *n* = 2067) (**A**), participants by sex (total *n* = 715,539) (**B**), participants-events ratio (**C**) and participants by Age group (**D**) of 100-km ultra-marathons from 1960 to 2019.

**Figure 2 ijerph-18-00362-f002:**
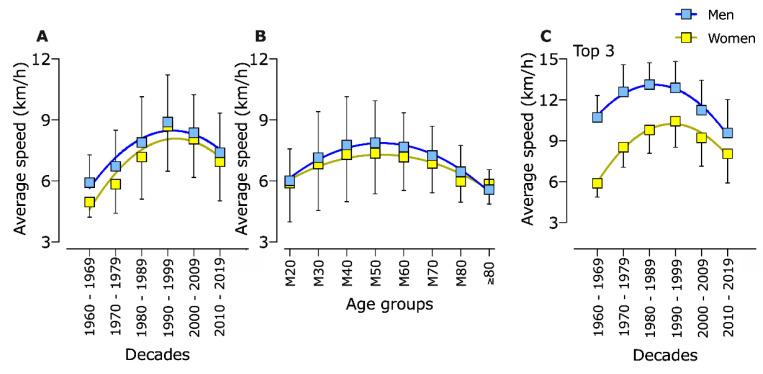
Average running speed of men and women participating in 100 km ultra-marathons across decades (**A**), by age groups (**B**), and considering only top three athletes from each event (**C**).

**Figure 3 ijerph-18-00362-f003:**
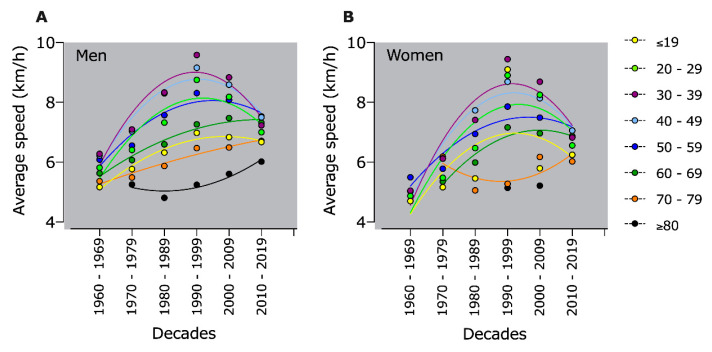
Average running speed in 100-km ultra-marathons by age groups of men (**A**) and women (**B**).

**Figure 4 ijerph-18-00362-f004:**
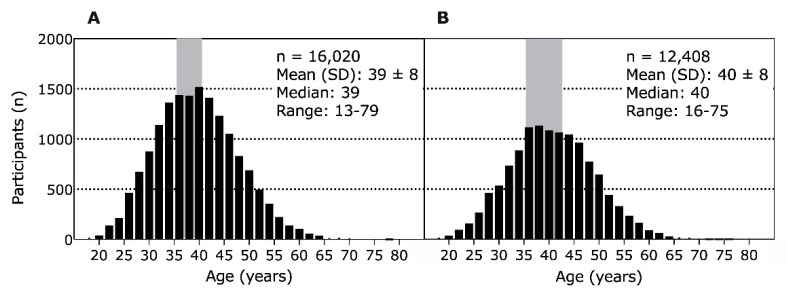
Histogram of top three athletes distributed by their age (participants × age). (**A**) men, (**B**) women.

**Table 1 ijerph-18-00362-t001:** Average running speed (km/h) of men and women running 100-km ultra-marathons by age group. Data are expressed as mean ± standard deviation (SD). The *p*-value displayed refers to post hoc analysis to detect between-group differences.

Age Group (Years)	Men Running Speed (km/h)	Women Running Speed (km/h)	*p*-Value
20–29	6.0 ± 1.6	5.9 ± 1.9	0.360
30–39	7.2 ± 2.3	6.8 ± 2.3	<0.001
40–49	7.8 ± 2.4	7.3 ± 2.3	<0.001
50–59	7.9 ± 2.1	7.4 ± 2.0	<0.001
60–69	7.7 ± 1.7	7.2 ± 1.6	<0.001
70–79	7.3 ± 1.4	6.9 ± 1.4	<0.001
80–89	6.5 ± 1.3	6.0 ± 1.0	<0.001
90–99	5.6 ± 1.0	5.9 ± 1.0	0.568

**Table 2 ijerph-18-00362-t002:** Average running speed (km/h) of men and women finishing top three in 100-km ultra-marathons by age group. Data are expressed as mean ± standard deviation. The *p*-value displayed refers to post hoc analysis to detect between-group differences.

Top 3			
Age Group (Years)	Men Running Speed (km/h)	Women Running Speed (km/h)	*p*-Value
20–29	9.2 ± 2.2	8.2 ± 3.0	0.130
30–39	10.3 ± 2.8	8.4 ± 2.5	<0.001
40–49	10.8 ± 2.8	8.7 ± 2.4	<0.001
50–59	10.4 ± 2.5	8.6 ± 2.1	<0.001
60–69	9.5 ± 2.1	8.1 ± 1.9	<0.001
70–79	8.4 ± 1.8	7.6 ± 1.8	0.002
80–89	6.7 ± 1.5	7.0 ± 2.3	0.821

## Data Availability

Publicly available datasets were analyzed in this study. This data can be found here: https://statistik.d-u-v.org/geteventlist.php (accessed on 19 February 2020).

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
