# Peer review of "An Analysis of Participation and Performance of 2067 100-km Ultra-Marathons Worldwide"

_ijerph, 2021, doi:10.3390/ijerph18020362_

Round 1

Reviewer 1 Report

Dear authors,

I would like to congratulate you for the provided changes to integrate all reviewers’ comments. Although the scientific of the present work might not seem so obvious given the descriptive nature and employed methods, authors have proven the ability to argument and answer to most of the reviewers’ suggestions.

The present manuscript is a clear and enhanced version of the previous one. Particularly, major changes were made in the Discussion section, according to the reviewers’ comments. Suggestions for future studies are now addressed in the conclusions part, while the present study limitations are shortly recognized. Performance gap disappeared in older age groups possible explanations are now presented using recent findings, providing clear implications for practice. Results found that older athletes get faster in last decades and this question has now been discussed according to the athletes sporting background, VO2max and muscle mass development throughout lifespan in endurance athletes. A new sub-section was included (Younger and older women achieve a similar performance to their male counterparts) according to a reviewer suggestion. I am pleased with the contribution for this version of the manuscript, particularly after reading the discussion section. Nevertheless, I have some final remarks:

Line 191 and 195: in accordance with previous results, please include effect size results in Table 1 and Table 2.

Line 263: “56 years”.

Author Response

Reviewer 1

Dear authors,

I would like to congratulate you for the provided changes to integrate all reviewers’ comments. Although the scientific of the present work might not seem so obvious given the descriptive nature and employed methods, authors have proven the ability to argument and answer to most of the reviewers’ suggestions.

The present manuscript is a clear and enhanced version of the previous one. Particularly, major changes were made in the Discussion section, according to the reviewers’ comments. Suggestions for future studies are now addressed in the conclusions part, while the present study limitations are shortly recognized. Performance gap disappeared in older age groups possible explanations are now presented using recent findings, providing clear implications for practice. Results found that older athletes get faster in last decades and this question has now been discussed according to the athletes sporting background, VO2max and muscle mass development throughout lifespan in endurance athletes. A new sub-section was included (Younger and older women achieve a similar performance to their male counterparts) according to a reviewer suggestion. I am pleased with the contribution for this version of the manuscript, particularly after reading the discussion section. Nevertheless, I have some final remarks:

Answer: We thank the reviewer for the words of appreciation and value for incorporating the reviewers' suggestions for improvement.

Line 191 and 195: in accordance with previous results, please include effect size results in Table 1 and Table 2.

Answer: We thank the reviewer for this critical comment and have added the corresponding effect sizes to Tables 1 and 2 for clarity.

Line 263: “56 years”.

Answer: We thank the reviewer for the formal correction and change it.

Reviewer 2 Report

Dear authors, there has been an improvement in the article but there are still aspects that need to be modified. They are the following:

The line number must be indicated for each response to the reviewers as indicated in the rules of the journal.

Abstract: The abstract should be a total of about 200 words maximum.

There are 355 words in the abstract that exceed the limit set by the journal.

2.2. Data Sampling

This way of referring to websites is not appropriate. Please check the rules of the journal.

Line: 123 - (http.://statistik.d-u-v.org/geteventlist.php).

Line 126: Misprint -  Quotes ”

Results

Line 152: mistake - Figure1-D

Data sampling

I reiterate the need to indicate free access to the data in a specific manner. That it is public does not mean that it is free (e.g. municipal sports services).

The strength of this research is the sample, both of the total number of events and of subjects. However, the hypothesis raised "i) the number of participants worldwide in 100-km" does not involve any scientific effort, the final result being quite obvious. I suggest that it be revised by implementing some other variable such as: relation events-participants-edition. And finally, compare them at least.

This comment is not answered correctly.
It is not easy to find it in the paper because the line is not indicated. In addition, it is not rectified in the section that they indicate.

I also highlighted the importance of the relation events-participants-edition NOT participants-event.

Finally, they indicate that it is added as a hypothesis and appears directly in Discussion. It must be included as a hypothesis if you present it.

There is an error in the numbering of sections:

4.2. Ratio participants/event - Line 242

4.2. Performance gap disappeared in older age groups - Line 255

I reiterate: The hypotheses are not demonstrated with any structural model.

Example:

Managerial implication

It is advisable to add this section independently.

Limitations and Future Research Lines

It is advisable to add this section independently.

References

No journal presents the abbreviated form as indicated in the journal's regulations.

Line 445 - DUV: The full name must be given.

Author Response

Reviewer 2

Dear authors, there has been an improvement in the article but there are still aspects that need to be modified. They are the following:

The line number must be indicated for each response to the reviewers as indicated in the rules of the journal.

Answer: We thank the reviewer for this advice and include the line references in the further comments.

Abstract: The abstract should be a total of about 200 words maximum.

There are 355 words in the abstract that exceed the limit set by the journal.

Answer: We thank the reviewer for noting a shorter abstract and have shortened it significantly (lines 21-25).

2.2. Data Sampling

This way of referring to websites is not appropriate. Please check the rules of the journal.

Line: 123 - (http.://statistik.d-u-v.org/geteventlist.php).

Answer: We are pleased to note the criticism and have reviewed the rules of the journal. The 3rd section Structure and Formatting (https://www.mdpi.com/authors/layout#_bookmark6) states that the main text of an article can be supplemented by additional documents or sections. URLs can be used to refer to the data. For this reason, we have left the URL specification in the text line 134.

Line 126: Misprint -  Quotes ”

Answer: We thank the reviewer for the formal correction and change it.

Results

Line 152: mistake - Figure1-D

Answer: We thank the reviewer for the formal correction and change it.

Data sampling

I reiterate the need to indicate free access to the data in a specific manner. That it is public does not mean that it is free (e.g. municipal sports services).

Answer: In its data protection declaration (https://statistik.d-u-v.org/dataprivacy-en.php), this website provides detailed information on the collection of personal data and refers to the Federal Data Protection Act and Telemedia Act. These personal data come from publicly available sources and is therefore a secondary use of the “Deutsche Ultramarathon Vereinigung” (DUV). DUV also obtains its data directly from the websites. Our data collection was also collected directly from the website, but not personal data, but anonymous data. Accordingly, there is no need to obtain permission or the obligation to provide information to the DUV, as the data are publicly accessible and are therefore considered to be third-party use.

The strength of this research is the sample, both of the total number of events and of subjects. However, the hypothesis raised "i) the number of participants worldwide in 100-km" does not involve any scientific effort, the final result being quite obvious. I suggest that it be revised by implementing some other variable such as: relation events-participants-edition. And finally, compare them at least.

This comment is not answered correctly.
It is not easy to find it in the paper because the line is not indicated. In addition, it is not rectified in the section that they indicate.

Answer: We thank the reviewers for pointing out this discrepancy and would like to clarify it. As part of your first correction, we added the aims in lines 95-96 "Therefore, the aim of this study was.....,second, to put the increasing number of participants in relation to the increasing number of event." Unfortunately, although we did not implement your suggestion in your way , we will propose and include it for future studies. The newly introduced participant/event ratio was also added to the results after the first correction in line 100-101: "(ii) the participants/event ratio would be declining". Furthermore, the hypothesis in line 198 " ...secondly, that the ratio participants/event decreased in the last decades." and the result in line200-202 "The main results were....(ii) the ratio participants/event reaches a plateau in last decade after a decline,." Thus, in Section 4.2 Ratio participants/event, from line 238-250, a separate section was devoted to this topic: "In contrast to the increasing number of participants worldwide, a decline of participation in particular single events could be observed [3,9,30]. For example, the decrease of participants in single events can be shown in the oldest 100-km ultra-marathon in the world, the “100km Lauf Biel” [3], or in the “Comrades Marathon” [30]. This decline of participants in particular single events could be a result of an increasing number of events. An increase in events could also be shown in other long-distance events such as the marathons in China [33]. In our study, such a strong increase of 100-km ultra-marathons events could be demonstrated, especially in the last decade from 2010-2019. In this last decade of an increasing number of events and increasing participation, the ratio between participants and events was stable. In previous decades up to 1989, the ratio between participants and events decreased. This leads to the assumption that in previous decades before 1989 there was a redistribution of the participants to the events. So, the number of participants in single events declined as the number of events offered increased

I also highlighted the importance of the relation events-participants-edition NOT participants-event.

Answer: We thank the reviewer for his excellent comments and we consider them for a future analysis where we explicitly will investigate this aspect. As mentioned below, during the period 1960-2019, only the 100 km Lauf Biel in Switzerland was held without interruption. During this period of 60 years many races were held but also many races disappeared. The advantage of the database of DUV is the fact that the persons there ask all races for historic race results printed on paper and transfer then in their data base

Finally, they indicate that it is added as a hypothesis and appears directly in Discussion. It must be included as a hypothesis if you present it.

Answer: We thank the reviewer for this critical objection and have indicated the correct details of the revision -as already mentioned above. Thereby, the hypothesis and the main findings were adjusted in the introduction as well as in the discussion in the introductory part. In the following section 4.2.-inserted above- this is then considered in more detail.

There is an error in the numbering of sections:

4.2. Ratio participants/event - Line 242

4.2. Performance gap disappeared in older age groups - Line 255

Answer: We thank the reviewer for pointing out the incorrect enumeration and correct this.

I reiterate: The hypotheses are not demonstrated with any structural model.

Example:

Answer: We thank the reviewer for this excellent suggestion to use a structural equation model for the statistical analyses. Thereby, the hope would be to establish a dependency relationship between the different variables. This could especially support our presented trend of performance gaps in age and sex. Unfortunately, we could not implement this structural equation model because there is only one 100km ultramarathon event that has consistently taken place in the time span from 1960 to 2019. This historical 100km ultramarathon race takes place in Biel, Switzerland, and has already been analyzed in the study "Participation and performance trends in the oldest 100-km ultramarathon in the world” (doi:10.3390/ijerph17051719). In our study, however, all events that took place in the time span 1960-2019 were examined. But in this time span events in the ultramarathon scene come and go. There is no temporal continuity of events for this time span. This is not uncommon for such long distances and can be seen even in the popular sport " Marathon". In the case of the marathon, there are also only a very few major events in the world, such as Boston and New York City. Thus, in a structural equation model, only the 100km ultramarathon event in Biel could be included in our study, whereas all other races could not be included. But this high number of events makes the strength of our study with a trend analysis possible. Our study can certainly reveal trends that can be used in subsequent studies to develop a structural equation model. A possible study design would then analyze a time period in which there is continuity of events over time, such as events only from the last decade. By our trend analysis we unfortunately accept that complex correlations of the variables remain undiscovered. This gap will surely be closed by new studies due to increasing data size and rising popularity of ultramarathons.

For better classification and also justification of the very simple conclusion of the increasing number of participants worldwide, we have added another study and a further explanation of our study aims in the section Introduction. This should help the reader to understand the special event situation of ultramarathons.

Lines 68-83:

However, two studies on participation analysis of ultra-marathons showed that there was a decline in the number of participants in these individual events. The number of participants in single ultra-marathons like the “100km Lauf von Biel” as the oldest 100-km ultra-marathon in the world [9] or the “Comrades Marathon” primarily increased in the 1970s and 1980s. Thereafter a continuous decline occurred [9, 30]. The initial increase and the subsequent decrease of participation depended primarily on the number of male participants [9]. However, the worldwide trend of 100km ultramarathons from 1998-2011 shows an increasing number of participants [8]. This contrary development of the participants in ultra-marathons in particular single events and the overall view worldwide highlights the importance of examining participants worldwide with a more extensive database to better understand the trends of athletes running ultra-marathons. Additionally, there is currently only data on the number of participants for other distances and regions such as the 161-km (100miles) ultra-marathons in North America [1]. The available information on the development of the number of participants in 100-km ultra-marathon is limited to a short period of investigation from 1998-2011 [8]. This study aimed to close this gap by a worldwide data collection of participation in 100-km ultra-marathons over several decades. Furthermore, our study aims to find out the worldwide trend of 100km ultramarathon participation despite changing organizers and events.

Managerial implication

It is advisable to add this section independently.

Answer: Because measures should primarily be taken with regard to future research topics, we have added the managerial implication under the section Limitations and Future research lines. Because from the present trend analyses so far only indications can be drawn e.g. for improvement in medical fields or training plans of older athletes. Also, we have included your important suggestion to analyze an event/participant ratio in this section to emphasize the importance of this issue and to highlight it for future studies.

Lines 371-383:

“Another limitation of the study is the lack of continuity of events, so that in the data collection events sometimes occur only for short periods of time. This makes it difficult to establish a structural equation model to identify multifactorial dependency relationships. Establishing an event/participant ratio is also difficult because only one event occurred permanently during the period of data analysis. The other events passed and failed during the period of data analysis. Implications for future studies would therefore be to analyze the event/participant ratio within a shorter period of time, e.g., the next decade.

Further measures should be taken with regard to the growing older age groups, as with the higher age exists the risk of medical events with the so far existing standards. Adapted medical assessment for the older athletes would be useful in avoiding medical events such as sudden cardiac death. For coaches and older athletes preparing for an ultramarathon, specific training plans and nutrition plans for the elderly athletes' requirements are of new importance as well and should be newly developed and adapted for this older age group.”

Limitations and Future Research Lines

It is advisable to add this section independently.

Answer: We thank the reviewer for the suggestion for improvement and add a new paragraph 5. enumerating from line 389 to 297 the limitations and future research lines:

«Future studies should include more participant characteristics such as aerobic fitness per unit of body, muscle mass, fat mass or years of training. Despite the large amount of data available on the DUV website, these characteristics cannot be retrospectively recorded in our study. Therefore, the gap in the study can only be closed by using a different study design to identify the origin of the data observed in our study in the older age group of ultra-marathon athletes. Another strength of this study is the low number of participants in the higher age groups despite the high volume of data. This showed a trend between age and performance, but the result could be overestimated in the higher age groups.»

References

No journal presents the abbreviated form as indicated in the journal's regulations.

Line 445 - DUV: The full name must be given.

Answer: We thank the reviewer for this advice and spell out the Abbreviation.

Reviewer 3 Report

Congratulations to the authors. This is an original study and the results are very interesting. The objective of the paper was analyze the number of successful finishers and the performance of 16 the athletes in the 100-km ultra-marathons worldwide. Also, the authors deal with an interesting and important topic from a scientific and practical point of view, and the study is well designed. In this line, there are only minor comments that should be addressed:

I present the following in the order which the paper was written :

Introduction

Provides enough information to understand the issue raised in this manuscript. Nice job.

Materials and Methods

- EKSG 01-06-2010 this is the Project identification code? If not, please include it.

- Methods are well described. Did you perform some type of statistical power calculation; at least a-posteriori?

Results

- line 159: it would be interesting to include a space in sexxdecade (sex x decade) and groupxdecade (group x decade).

Discussion

- I found it very complete, this is due to the changes that the authors have made with respect to the comments of the rest of the reviewers.

- Line 263: Please insert a space after 56.

Conclusion

- It would be interesting to include some limitations of the study.

Author Response

Reviewer 3

Congratulations to the authors. This is an original study and the results are very interesting. The objective of the paper was analyze the number of successful finishers and the performance of 16 the athletes in the 100-km ultra-marathons worldwide. Also, the authors deal with an interesting and important topic from a scientific and practical point of view, and the study is well designed. In this line, there are only minor comments that should be addressed:

I present the following in the order which the paper was written :

Introduction

Provides enough information to understand the issue raised in this manuscript. Nice job.

Answer: We thank the reviewer for the good criticism.

Materials and Methods

EKSG 01-06-2010 this is the Project identification code? If not, please include it.

Answer: We thank the reviewer for pointing out this unclear abbreviation and fully attribute it to “Ethical Committee St.Gallen”

- Methods are well described. Did you perform some type of statistical power calculation; at least a-posteriori?

Answer: We appreciate the reviewer's comment. Post-hoc power analyses were conducted. Due to the high sample size in all models (n > 400,000), all ANOVA models had: 1 - beta = 1.0.

Results

- line 159: it would be interesting to include a space in sexxdecade (sex x decade) and groupxdecade (group x decade).

Answer: We thank the reviewer for the formal correction and change it.

Discussion

- I found it very complete, this is due to the changes that the authors have made with respect to the comments of the rest of the reviewers.

Answer: We thank the reviewer for the well criticism.

- Line 263: Please insert a space after 56.

Answer: We thank the reviewer for the formal correction and change it.

Conclusion

- It would be interesting to include some limitations of the study.

Answer: We thank the reviewer for pointing out the insufficient enumeration of weaknesses in this study. One weakness already mentioned in line 391 is certainly the retrospective analysis of existing data, so that no specific data on training or physiological function of the participants were available. Another limitation of our study for added to the Conclusion as follows: Line 395-397: “Another strength of this study is the low number of participants in the higher age groups despite the high volume of data. This showed a trend between age and performance, but the result could be overestimated in the higher age groups.”

Round 2

Reviewer 2 Report

Dear authors, there has been an improvement in the article but there are still aspects that need to be modified. They are the following:

Abstract: The abstract should be a total of about 200 words maximum.

There are 245 words in the abstract that exceed the limit set by the journal.

References

No journal presents the abbreviated form as indicated in the journal's regulations.

Congratulations on the corrections and clarifications made.

Author Response

The abstract is now shortened as requested

This manuscript is a resubmission of an earlier submission. The following is a list of the peer review reports and author responses from that submission.

Round 1

Reviewer 1 Report

Abstract

The abstract adequately summarizes the research by collecting the objective, the sample and the results offer the main findings.
However, the methodology used and the procedure for data collection are not specified.

Keywords:
The selected Keywords are suitable and will allow a quick search by interested researchers. I would add "male-female differences" to the list to facilitate researchers' searches.

Introduction
The general idea of the research is presented with an adequate and updated theoretical framework. It first contextualizes the sports event and its technical characteristics and then adequately summarizes the different findings in the area of performance. The introduction ends with three hypotheses.
The referenced research is from recent years.

Materials and Methods
The selected sampling model is not described.

Data sampling
It would be necessary to indicate whether Deutsche Ultramarathon Vereinigung has given permission to use your data or they are free. In this case, please let us know if this organisation has been informed of the study and what its position is.

The strength of this research is the sample, both of the total number of events and of subjects. However, the hypothesis raised "i) the number of participants worldwide in 100-km" does not involve any scientific effort, the final result being quite obvious. I suggest that it be revised by implementing some other variable such as: relation events-participants-edition. And finally, compare them at least.

On the contrary, the hypotheses "(ii) there were no further significant performance gaps in age groups 163 over 60 years or older", (iii) older age groups over 70 years increased their performance the last 164 decades" are interesting but I think it is necessary to show in tables the results (by decades, every 20 years, etc.). 

Results

I strongly advise that the charts in this section be placed within this section and it is not a discussion.
The hypotheses are not demonstrated with any structural model.
The analysis of the results is scarce and merely descriptive.

Discussion
It is necessary to remove the figures that do not correspond directly to this section. The references with which you contrast the ideas are appropriate.

Conclusions
They respond to the hypotheses raised.

Strengths and Limitations
It does not raise any.

Future Lines of Research
Not included.

I believe that it does not respond to the aim and scope requested in the International Journal of Environmental Research and Public Health (IJERPH).

The article is interesting in the field of sports management, however, I do not find any link with the aim and scope of this journal.

Line 47 - misprint: "35years
Figure 1-d "Agegroups"
Figure 2- "Agegroups"

Reviewer 2 Report

I read the manuscript with interest, but I have several concerns.

  1. The hypotheses seem to be constructed post-hoc. Why this particular distance of an ultramarathon was chosen out of a database reporting all ultramarathon distances?
  2. Most of the results presented can be found on the DUV website (including the graphs), but without statistical analysis, which limits the novelty of the findings.
  3. Ultra-marathon in a race longer than 42 km and not lasting longer than 6 hours. One can run 30 km in longer than 6 hours, which is obviously not considered as an ultramarathon.
  4. The results are rather intuitive. Every sports medicine doctor (interested in this topic) sees on a daily basis that there are more ultramarathon participants in recent years (and more races). The age of participants is increasing, but because of the larger number of participants and increasing age the performance is not increasing anymore. The interesting findings include sex differences and the appearance and performance characteristics of older athletes (including 80 years old + participants). 

Reviewer 3 Report

Dear authors,

I would like to congratulate you on the I would like to thank the authors for the effort to carry out this descriptive study. In my opinion the overall appreciation of the manuscript is good, considering the rationale and framework for the research question. All the provided elements throughout the manuscript are clear and informative for the potential reader. Nonetheless, I have some concerns or comments for your consideration, as following:

Abstract

Line 27 to 29: I find it odd the inclusion of this argument in the summary, once it was not properly explored or discussed throughout the manuscript. I strongly suggest that authors revise the discussion section and include practical implications, as suggested in the abstract.

Introduction

I find it well written, providing the potential reader with a proper framework for understanding the problematic around the study and research question. I find it particularly interesting as the authors point out the gaps in literature, as the rationale for the present study is presented. However, it can be improved by a more fluent narrative.

Line 51 to 55: “different findings …”, however only one study was cited. Also, how can this study be illustrative of a general trend for a decline in ultra-marathon participants?

Line 56: Again, this statement is based in one single event. Can it be generalized and consider a worldwide trend? Please improve the flow between paragraphs as this idea is explored.

Line 77: “as well” as. Please correct this.

Line 78 and 79: the sentence is not clear. Please reformulate. I would recommend authors not the state “a further aim is to…” as this might be confusing, Also, objectives are clearly stated in lines 90 to 100.

Line 80: please remove the first sentence as the hypothesis are clearly stated in lines 90 to 100.

Material and Methods

Line 119: I believe the correct tense is “were”. Please correct.

Line 117: whenever possible, I would suggest authors to include more information regarding the effect size and corresponding thresholds interpretation.

Results

All outcomes are appropriate for the study design and the graphic elements included in the manuscript allow the potential reader to have a clear and complementary interpretation of the observed results.

Discussion

This section is organized by topics, providing the potential reader an objective view about the key points of the present study. However, when analyzing and discussing the observed results, practical implications for coaches and athletes, should be explored and highlighted. Also, limitations should be adequately recognized and suggestions for future studies presented.

Line 229: other possible explanations should be included and further discussed. For example, aerobic fitness expressed per unit of body mass was an important variable to distinguish ultra-trail runners by competitive level (Oliveira-Rosado et al., 2020). In addition, see Laurin et al. (2019) for how lifelong endurance/aerobic type exercise can preserve muscle mass and function with age. This information can enhance our understanding about performance characteristics in long-marathon participants and practical implications suggested.

Line 267: the benefits of physical activity and exercise throughout life span could also be emphasized as master athlete’s performance should not be considered as an exception but as an indicator on physical function and health (Valenzuela et al., 2019).

Conclusion

Practical implications must be emphasized even if the present study has a descriptive nature.

References

Line 341: Is it missing some information? If so, please include it.